# Beef Steers and Enteric Methane: Reducing Emissions by Managing Forage Diet Fiber Content

**DOI:** 10.3390/ani13071177

**Published:** 2023-03-28

**Authors:** Daniel Santander, Juan Clariget, Georgget Banchero, Fabiano Alecrim, Claudia Simon Zinno, Julieta Mariotta, José Gere, Verónica S. Ciganda

**Affiliations:** 1Instituto Nacional de Investigación Agropecuaria, Estación Experimental La Estanzuela, Ruta 50 km 11, Semillero, Colonia 70006, Uruguay; 2Departamento de Geoquímica, Universidade Federal Fluminense, Outeiro São João Baptista s/n, Niterói 24020-141, Brazil; 3Engineering Research and Development Division, National Technological University (UTN), National Scientific and Technical Research Council (CONICET), Buenos Aires C1179, Argentina

**Keywords:** cattle, mitigation, intake, forage, sulfur hexafluoride, quality

## Abstract

**Simple Summary:**

Methane (CH_4_) is one of the main GHGs that is emitted by ruminant production systems, making its quantification important. A total of two forage diets with different fiber contents were fed to beef steers, and their enteric CH_4_ emissions were evaluated. Additionally, the predicted values from different models were compared with the observations from different studies. The animals with a moderate fiber diet had better performance and lower intensity of CH_4_ emissions. The model with the best accuracy and precision was the one that was suggested by the IPCC 2006 Guidelines.

**Abstract:**

Understanding the methane (CH_4_) emissions that are produced by enteric fermentation is one of the main problems to be solved for livestock, due to their GHG effects. These emissions are affected by the quantity and quality of their diets, thus, it is key to accurately define the intake and fiber content (NDF) of these forage diets. On the other hand, different emission prediction equations have been developed; however, there are scarce and uncertain results regarding their evaluation of the emissions that have been observed in forage diets. Therefore, the objectives of this study were to evaluate the effect of the NDF content of a forage diet on CH_4_ enteric emissions, and to evaluate the ability of models to predict the emissions from the animals that are consuming these forage diets. In total, thirty-six Angus steers ($\bar{x}$
= 437 kg live weight) aged 18 months, blocked by live weight and placed in three automated feeding pens, were used to measure the enteric CH_4_. The animals were randomly assigned to two forage diets (n = 18), with moderate (<50%, MF) and high (>50%, HF) NDF contents. Their dry matter intake was recorded individually, and the CH_4_ emissions were measured using the SF_6_ tracer gas technique. For the model evaluation, six prediction equations were compared with 29 studies (*n* = 97 observations), analyzing the accuracy and precision of their estimates. The emission intensities per unit of DMI, per ADG, and per gross energy intake were significantly lower (*p* < 0.05) in the animals consuming the MF diet than in the animals consuming the HF diet (21.7 vs. 23.7 g CH_4_/kg DMI, 342 vs. 660 g CH_4_/kg ADG, and 6.7% vs. 7.5%, respectively), but there were no differences in the absolute emissions (*p* > 0.05). The best performing model was the IPCC 2006 model (*r^2^* = 0.7, RMSE = 74.04). These results show that reducing the NDF content of a forage diet by at least 10% (52 g/kg DM) reduces the intensity of the g CH_4_/kg DMI by up to 8%, and that of the g CH_4_/kg ADG by almost half. The use of the IPCC 2006 model is suitable for estimating the CH_4_ emissions from animals consuming forage-based diets.

## 1. Introduction

Consumers worldwide have shown great concern about climate change and its consequences, especially those due to the greenhouse gas (GHG) emissions that are associated with animal and agricultural production systems. These systems produce approximately 17% of the global anthropogenic GHG emissions, and methane (CH_4_) and nitrous oxide (N_2_O) are the most commonly emitted gasses [1]. In Uruguay, the livestock sector is identified as one of the anthropogenic activities that contributes to GHG emissions the most, contributing 70% of the total emissions. Most of these are CH_4_ gas emissions, with enteric fermentation being the main contributor (46%) [2]. In addition, the enteric production of CH_4_ gas by cattle is a loss of energy from the feed consumed, resulting in a productive inefficiency [3,4,5]. Thus, improving the environmental efficiency of beef production systems would contribute to mitigating these GHG emissions and thus climate change [6].

The production of CH_4_ (methanogenesis) in ruminants is a by-product of the rumen (~90%) and large intestine (~10%) digestion that is derived from the fermentation of consumed carbohydrates [7,8]. In recent decades, scientists have made efforts to understand the factors affecting methanogenesis in the rumen, evaluating effects from animal genetics to livestock management techniques [6]. However, nutritional quality and the amount of feed intake are the main regulatory factors that have been identified which determine CH_4_ production [9]. In Uruguay, beef cattle systems are mainly based on diets with a high forage inclusion (>70%) [10] that show great variations in their nutritional quality [11]. Frequently, such a variation in quality results from the proportion of the structural carbohydrates that are present in the diet, which are expressed as components of neutral detergent fiber (NDF). The variability in the NDF content is mainly due to the plant species that are present in the forage diet, the maturity stage of such species, the pasture management, and the climate conditions [12,13]. The NDF content in climate template forages has been reported to vary from 35 to 55% in cultivated forages and between 50 and 80% in natural forages [11,14].

It is known that low quality diets with a high NDF content may favor and stimulate the methanogenic processes in the rumen [15,16]. However, studies that have aimed to explain how the increment in NDF intake affects the CH_4_ emissions have come to no clear conclusions. For example, Primavesi et al. [17] compared the CH_4_ emissions from animals that were fed with two sugarcane varieties with contrasting NDF values (44.2% and 54.7% NDF) and found that, despite no difference in the intake, the use of the lower NDF variety produced 30% less CH_4_ emissions than the higher NDF diet. On the other hand, Hammond et al. [18] evaluated the addition of NDF to chopped straw and soy hulls, (+47 g/kg DM) in two types of forage diets (corn silage and grass silage), finding no difference in the intake but also no difference in the methane emissions. Both studies included supplementation with different feeds, and this could be hiding the actual effects of reducing the NDF content on the measured variables.

The effects of diet quality on CH_4_ production also vary as a function of the quantity of the feed intake [16,19,20]. A higher feed intake results in higher CH_4_ emissions, mainly due to the increment in the amount of feed that is available for fermentation. This relation was shown by Boadi and Wittenberg [21], who evaluated the effects of cattle breed type and forage quality on the methane production under ad libitum, and restricted the feeding conditions. In both situations, the level of feed that was offered affected the CH_4_ production, and the dry matter intake (DMI) was strongly correlated with methane production (*r*^2^ = 0.8).

The individual feed intake of the forage under grazing conditions appears to be difficult to determine. In this sense, indirect methods of intake measurement, such as external markers, can generate uncertainties that, in turn, prevent the measurement of accurate results [22,23]. On the other hand, the use of automatic feeders, in which cut forage is offered to cattle, is an effective alternative to avoid this problem and could be considered to be an accurate tool to measure individual intake [4,24]. Therefore, the use of such techniques could give precise DMI results and improve our understanding of the effect of the NDF content of forage diets on enteric CH_4_ emissions.

In recent years, different prediction equations (Tier 2 approach) have been developed as a strategy to help estimate the methane emissions from livestock [25,26,27,28]. Using different dietary variables to estimate the enteric CH_4_ production, these equations have shown a good accuracy. A set of them has been developed using the NDF variable as one of its main components [25,29]. Furthermore, Escobar-Bahamondes et al. [27] and Benauda et al. [30] evaluated the performance of such prediction equations, finding correlation coefficients of up to 0.6 and 0.5 for the diets with a high forage inclusion and for the diets with different NDF qualities, respectively. Although the validation of these equations for diets with a high forage inclusion and different NDF contents is still limited, they are a particularly useful and promising approach to defining CH_4_ emissions while considering the NDF diet content.

In the present study, we hypothesize that a forage diet with a lower NDF content would reduce the methane emissions and the intensity of these emissions, when compared with the same forage diet with a higher NDF content. Therefore, the objective of this study was to evaluate the effect of the NDF content of a forage diet on the enteric CH_4_ emissions of cattle growing steers. To achieve this objective, the following specific objectives were proposed: (i). to identify the effect of increasing the fiber content (NDF) on the intake, digestibility, average daily gain, absolute enteric CH_4_ emissions, and emission intensity; and (ii). to evaluate the predictive power of the different enteric CH_4_ prediction equations that have been proposed for feeding forage diets and NDF.

## 2. Materials and Methods

The study was conducted at the Experimental Station INIA La Estanzuela (Colonia, Uruguay, GPS Coordinates: S Latitude 34°20′ 23,72′′ S, W length 57°41′ 39,48′′ W) during the months of April to July 2021 (97 days). All the procedures involving animals were approved by the Committee for the Ethical Use of Animals at the National Institute of Agricultural Research (INIA-Uruguay, Protocol number 2020-5).

### 2.1. Experimental Design and Animals

The study was conducted using 36 18-month-old Angus steers with an average live weight of 437 ± 7 kg (BW). The study was carried out between April and July (97 days) of 2021. The animals were blocked by BW and randomly assigned to one of two treatment groups (n = 18). The steers were distributed into three different pens, where each pen had four automatic feeders (INTERGADO, Minas Gerais, Brazil) with access-limiting doors. Before beginning the trial, the animals were subjected to a 47-day common feeding period on the automatic feeders, which consisted of ad libitum grass haylage. This acclimatization period was used to accustom the animals to their environment and feeding system, and to reduce any latent influence of their previous nutritional management. The daily intake for each steer was registered automatically using the ear tag of each steer and the electronic scales of each feeder. During the experiment, the corresponding treatment was offered ad libitum three times a day at 06:30 h, 13:00 h, and 19:30 h.

### 2.2. Treatments

The treatments consisted of two forage diets of different qualities: (1). The moderate-fiber (MF) forage diet, which consisted of 100% alfalfa and orchard-grass haylage; and (2). the high-fiber (HF) forage diet, which consisted of 70% alfalfa, plus orchard-grass haylage, and 30% barley straw. The complete chemical composition of both diets is shown in Table 1. To avoid selectivity of diet components by the animal, the forage was offered homogeneously. For this purpose, the differences in the fiber sizes (<=70 mm) were avoided, both at the time of the forage conservation (harvesting, airing, and preservation) and during the administration (chopping and mixing). The amount of feed that was offered was adjusted daily to guarantee a daily refusal of 5% of the total amount that was supplied, in order to ensure ad libitum intake. The feeders were cleaned, and the refuse was removed and discarded three times per week.

### 2.3. Chemical Analysis of Feed

The food samples were collected three times a week from the feeding pens, making monthly pools for the chemical analysis. The feed samples were dried in a forced-air oven at 60 °C for 48 h and were ground through a 1 mm screen before the chemical analysis. The dry matter (DM) concentration was determined by drying at 105 °C in an oven for 24 h (UNIT-ISO 6496:1999), and the ash content was determined by incineration at 600 °C for 4 h (UNIT-ISO 5984-2002). The crude protein (CP) was determined using the Kjedahl method (7.021 procedure), according to A.O.A.C [31]. The ether extract (EE) was determined with an ANKOM^xt15^ extractor, using petroleum ether extraction (AOCS AM 5-04, Ankom Technology Corp., Fairport, NY, USA). The NDF and ADF were determined with an Ankom Fiber Analyzer (ANKOM^a2000i^, Ankom Technology Corp., Fairport, NY), using the filter bag technique. The acid detergent lignin (Lig.) was determined using a sequential analysis and is expressed exclusively in terms of residual ash, according to the method of Van Soest et al. [32]. The gross energy (GE) was determined using an adiabatic bomb calorimeter (Gallenkamp Autobomb; Loughborough, Leics, UK). The concentration of the non-fiber carbohydrates (NFC) was calculated as suggested by the NRC [33] (Equation (1)).
NFC (%) = 100 − (NDF% + CP% + EE% + Ash%)(1)

### 2.4. Feed Intake and Animal Performance

The individual daily feed intake of each animal was recorded and monitored throughout the period by the automatic feeders (INTERGADO, Minas Gerais, Brazil). Each animal was identified with an electronic ear tag, which was linked to the door of each feeder and its corresponding diet. The steers’ LWs, without fasting, were recorded every 14 d. The weighing was performed before 0630 h, when the first meal was provided. The average daily gain (ADG) was calculated as a lineal regression of the LW (unshrunk) slope for each steer, as recommended by Crews and Carstens [34]. The daily intake of the OM, CP, NDF, ADF, Lig, Ash, EE, and GE were estimated for each steer by multiplying its individual intake by the content of each chemical component of the diets.

### 2.5. Digestibility of Diet and Fiber

For the determination of the apparent digestibility of the digestive tract, individual feces samples were extracted directly from the rectum. The samples were collected over five consecutive days [35], and at the end of the experimental period. They were air-dried at a temperature of 65 °C for 72–96 h. The samples were analyzed via the insoluble acid ash technique as an internal marker of digestibility (UNIT-ISO 5985-2002). In addition, an analysis of the digestibility of the NDF was carried out with the NDF content of the fecal samples, the total fecal excretion that was estimated from the previously analyzed apparent digestibility, and the individual intake of each animal.

### 2.6. Determination of CH_4_ Emissions

The determination of the enteric CH_4_ emissions was performed using the sulfur hexafluoride (SF_6_) tracer technique [16], adapted by Gere and Gratton [36]. A total of ten days before the beginning of the measurements, each animal was given an oral permeation tube that was filled with SF_6_, using a plastic dosing applicator. The permeation rates of the SF_6_ from the tubes were, on average, 6.57 mg/day. Burped and breath air samples were collected for five consecutive days before the end of the 97-day experimental period. The CH_4_ collection devices for each animal consisted of two stainless steel cylinders of 0.5 L, which had been previously cleaned with high-purity nitrogen gas (N_2_) and pre-evacuated (<0.5 mb). Both the cylinders were coupled to a muzzle and placed on each side of a backpack that was adjusted on the animal. Each cylinder was connected to an airflow regulator that was restricted by a steel ball bearing, which ended approximately 3 cm from the animal’s nostril. The inflow regulators were calibrated before each collection event, in order to allow a vacuum to remain in the steel cylinders of approximately 500 mb by the end of the sampling period (five days). In total, three additional cylinders were placed on each feeding pen in the experimental area to collect the air samples that represented the environmental CH_4_ and SF_6_ concentrations (background samples).

Following the procedure that was performed by other authors [37,38,39], at the end of the five-day sampling period, the cylinders were removed and the post-sampling pressure was measured. The containers with pressure values of 400–600 mb were considered to be valid and ensured good quality samples [36]. The containers with pressure values <400 or >600 mb were removed from the experiment. In total, 86% of the samples were considered to be valid for analysis. A total of four sub-samples were extracted from each container and stored in 6 mL vacutainers for determining the CH_4_ and SF_6_ concentrations using gas chromatography.

### 2.7. Gas Analysis and Calculation

The concentrations of the CH_4_ and SF_6_ were determined with chromatography. The sub-samples were analyzed using a gas chromatograph (Agilent 7890A, Santa Clara, CA, USA), with a flame ionization detector (FID) and an electron capture detector (ECD), for determining the CH_4_ and SF_6_ concentrations, respectively. The maximum delay between the collection and the determination of the CH_4_ and SF_6_ concentrations was 20 days. After conducting a chromatographic analysis of the samples, the emissions of the CH_4_ per animal were calculated using the permeation rate of each SF_6_ capsule (PR) and the atmospheric (atm) and enteric (ent) concentrations of the CH_4_ and SF_6_, considering the molecular weight (MW) of each one (Equation (2)).
CH_4_ (g day^−1^) = SF_6_ PR (mg day^−1^) × [CH_4_ ent − CH_4_ atm (ppm)/SF_6_ ent − SF_6_ atm] (ppt)] × [(16 (MW CH_4_))/(146 (MW SF_6_))] × 1000(2)

The methane conversion rate (Ym) was calculated following the equation that was proposed by the IPCC 2006 Guidelines [40], based on the conversion efficiency value of the gross energy intake (GEI) of the CH_4_ (Equation (3)).
Ym (%) = GEI (MJ/kg DM/day)/CH_4_ (g/day)(3)

### 2.8. Prediction Equation Models

#### 2.8.1. Database

A database of 97 observations (including our study) was used to assess the performance of the models predicting the CH_4_ emissions from ruminants. This database (Table A1, Appendix A) included 29 studies from 9 countries in 3 different continents. Studies from the Americas accounted for 80 of the observations from 24 of the studies, with Brazil being the main contributor. Europe accounted for 15 of the observations from 4 of the studies, and Asia accounted for 2 of the observations from 1 study. The enteric CH_4_ emissions that were included in this database were measured using respiration chambers (8/97, 8%), the SF_6_ tracer technique (75/97, 77%), and emission monitoring chambers (GreenFeed™, C-Lock Inc., Rapid City, SD, USA; 14/97, 15%). They were classified into two different production and measurement conditions: in confinement (20/97, 20%) and grazing (55/97, 80%).

#### 2.8.2. Data Pre-Processing

Data pre-processing was performed, as the collected data were sometimes incomplete (with missing values or variables of interest) or inconsistent (with different names or units for the same variable). The inconsistent data were corrected by using the same names and units across the studies. Some of the data on the gross energy content and the variables of the chemical composition that were not available were simulated using NRC 2016 software [41] to obtain them. The ranges of the variables that were chosen to be evaluated in the models were NDF (34–78%), forage (61–100%), GE (11.3–21.8 MJ/kg DM), digestibility (35–84%), body weight (233–712 kg), DMI (4.2–25.2 kg/day), and CH_4_ emissions (80–656 g/day).

#### 2.8.3. Selection Model

The CH_4_ emission estimation models that were chosen for this study were extracted from the work of Escobar-Bahamondes et al. [27] and Benauda et al. [30], based on forage diets and including the fiber variables in the equations (Table 2). Only the models with predictor variables or required information that were available in our database and experiment were selected. The IPCC 2006 tier 2 model [40] was chosen because it is frequently used in national inventories and scientific reports. The Ellis 2007 (a,b,c) models are described in [25], and the Moraes 2014 (H_AL) and (DL) models are described in [29]. The selected models are presented in the following table:

### 2.9. Statistical Analysis

All the animal and emission data were analyzed using Infostat 2020 software [42]. The normality test (Shapiro–Wilks test) was applied to all the variables. The model that was obtained was the following mixed model, Yijk = u + Ti + Pj + Bk + eijk, where Y is the dependent variable; u is the overall mean; T (i = moderate or high fiber) is the treatment effect, P (j = 1–3) is the pen effect, both as fixed effects; B (k = 1–12) corresponds to the block effect as a random effect; and e is the associated error. The means were compared with Fisher’s LSD test. The significance was declared as *p* ≤ 0.05, and trends were considered when *p* > 0.05 and *p* < 0.10.

The linear regression and Pearson’s correlation coefficients (*r*^2^) were calculated using R software (version 4.1.1) (R Foundation for Statistical Computing, Vienna, Austria) [43] to examine the relationship between the variables of the dry matter intake, NDF intake, and CH_4_ emissions. The significance was declared as *p* < 0.05, and trends were considered when *p* > 0.05 and <0.10.

For the analysis of the observed versus predicted emission models, the conclusions that were presented in Piñeiro et al. [44] were taken into account. R software (version 4.1.1) [43] with the “*metric*” and ”*biod3d*” packages was used, evaluating the linear regression, with Pearson’s coefficient (*r*^2^), root mean square error (RMSE), and deviation (RMSD) being used as the parameters for determining the effectiveness of the models at estimating the emissions of the CH_4_.

## 3. Results

### 3.1. Intake and Digestibility

The feed intake that was recorded for both groups of steers ranged from 1.8% to 2.25% of the BW (Table 3). The total DMI was higher for the MF diet compared with the HF diet (*p* < 0.05). For most components, except for ADF, the intake was higher in the MF than the HF diet (*p* < 0.05). In addition, the apparent digestibility (DMD), as well as the NDF digestibility (NDFD), was higher by ten percentage points when comparing the MF diet with the HF diet (*p* < 0.05).

### 3.2. Animal Performance

At the end of the experimental period, the average live weight was different between the two treatments (Table 4). It was observed that animals fed the MF diet weighed on average 41 kg of LW more than animals fed the HF diet (*p* < 0.05). Likewise, this difference is reflected in the difference that was presented in the average daily gain (ADG), which was twice as high in the MF group than in the HF group (*p* < 0.05).

### 3.3. Methane Emissions

The absolute enteric methane emissions (g CH_4_/day) did not show significant differences (*p* > 0.05) (Table 4). The methane emission intensity per kilogram of the dry matter intake (g CH_4_/kg DMI) was lower under the MF diet compared with the HF diet (*p* < 0.05). However, this difference was not present in the emission intensity that was evaluated per kg of the NDF consumed (*p* > 0.05). In the same sense, the emission intensity of the MF diet, which was expressed in relation to the average daily gain (g CH_4_/kg ADG), was almost half that of the HF diet (*p* < 0.05). Moreover, the methane conversion rate (Ym) was lower under the MF diet than under the HF diet (*p* < 0.05).

A linear regression analysis for the DMI variable and the absolute methane emissions (g/day), including all the observations that were obtained from our experiment, can be seen in Figure 1. A correlation value of *r*^2^ = 0.51 (*p* = 0.0043) was obtained. On the other hand, a linear regression analysis for the variables of the FDN intake and the absolute methane emissions (g CH_4_/day) showed a correlation value of *r*^2^ = 0.45 (*p* = 0.013), as can be seen in Figure 2.

### 3.4. Models’ Equations Evaluated

The six models that were selected and evaluated are presented in the observation versus prediction plots (Figure 3). The averages of the variables that were used in the evaluation of the equations were: NDF (50.5%), forage inclusion (91.8%), GE (17.63 MJ/kg DM), DMD (64.8%), BW (426 kg), DMI (11.18 kg DM/day), NDF intake (5.5 kg DM/day), and GEI (196.7 MJ/day). The mean of the observed emissions was 236 g CH_4_/day, while the means of the emissions of the prediction equations that were evaluated were: IPCC 2006 = 229 g CH_4_/day, Ellis 2007 = a, 265 g CH_4_/day, Ellis 2007 = b, 184 g CH_4_/day, Ellis 2007 = c, 197 g CH_4_/day, Moraes 2004 H-AL = 210 g CH_4_/day, and Moraes 2004 DL = 211 g CH_4_/day. The IPCC 2006 model obtained a lower RMSE (74.07 g CH_4_) and RSMD (128.29 g CH_4_). The best correlation was that of the IPCC 2006 model (*r*^2^ = 0.70, *p* < 0.001). The models with the lowest correlation were Ellis 2007 (a), and Ellis 2007 (b), with r^2^ = 0.28 for both. Ellis 2007 (a) and Ellis 2007 (b) presented the highest RMSE values of 118 and 131 g CH_4_, respectively.

## 4. Discussion

The present study explored how the manipulation of the fiber content (NDF) in a forage diet can affect its CH_4_ emissions and animal performance. The study focused on evaluating forage diets with high and moderate fiber contents (>, <50% NDF).

The chemical composition showed similarities with forages at a medium stage of maturity that are found in temperate climate zones (Table 1). Forage diets are characterized by a high proportion of structural carbohydrates in their composition [45]. As expected, the addition of fiber (+5% NDF, 52 g/kg DM) led to an increase in the fiber content of the diet, and consequently, a decrease in the other components (CP, EE) and in the non-structural/structural carbohydrate ratio in the HF diet. This ratio is used to define diet quality [46], and for this reason, we can define the MF diet as being of a better quality than the HF diet in terms of fiber.

The dry matter intakes (DMI) presented in Table 3 are similar to those reported for the same animal category fed forage-based diets [47]. The higher intakes that were observed from the animals consuming the MF diet (+20%) caused more of the chemical components of the diet to be ingested, when compared with the HF diet. Likewise, although the MF diet contained less NDF per kg of the DM than the HF diet, 0.49 vs. 0.54, respectively, the intake of this component by the animals was higher in the MF diet than in the HF diet. Regarding the ADF intake, similar results are presented between both the diets; however, the lower ADF content in the MF diet allowed the animals to obtain a higher intake of the DM, making it possible to observe the regulating effect of the ADF on the intake [48]. The intake of forage diets is mainly regulated by the rumen’s filling and distension, so that a higher amount of fiber content would result in a lower rate of degradation and less passage of feed through the rumen [45,49,50], thus causing a decrease in the intake in the animals that consumed the HF diet.

On the other hand, there were differences in the digestibility between both of the diets. The digestibility was 17% higher for the dry matter and 19% higher for the fiber in the MF diet compared with the HF diet (Table 3). These differences in the digestibility may lead to differences in the intake [51,52]. Assuming a linear relation between the marginal increase in the NDF digestibility and animal responses, Oba and Allen [53] proposed that, for each point decrease in the NDF digestibility, the intake decreased by 0.17 kg DM/day, which is very similar to the results of our study, with 0.19 kg DM/day. An increase in the retention times of the liquid phase of the feed in the rumen, and therefore a lower digestibility, led to a decrease in the animal intake [54]. Thus, Van Soest [45] proposed that the forage intake decreases when the digestibility is lower, and that this is related to the ratio between the soluble and structural carbohydrates (NFC:NDF) and lignin content. In our results, the NFC:NDF ratio was 0.43 vs. 0.35, and the lignin was 74 vs. 82 g/kg DM for the MF and HF groups, respectively. However, other studies have shown opposite results, as presented by Hammond et al. [55], where chopped barley straw and soybean hulls were added as fiber (+5% NDF) to a forage-based diet (corn or grass silage); however, this addition did not cause differences in the animal intake.

Another important point that could have affected the difference in the NDFD, DMD, and consequently, the intake (DMI), could be the levels of the crude protein intake, which was 0.5 kg DM/day higher for the MF diet than the HF diet (Table 3). Fiber degradability is strongly conditioned by the levels of ammonia (N-NH_3_) that are available in the rumen [56]. The microorganisms that are responsible for fiber degradation, especially cellulolytic microorganisms, require adequate levels of NH_3_ (endogenous or exogenous) and energy to act, which can affect the digestibility and intake of a forage diet [56]. Olson et al. [57] experimented with steers to evaluate the effects of two supplements at different proportions of starch and a degradable protein intake on the forage utilization and rumen function in a low-quality diet (tallgrass hay, 4.5% CP and 72% NDF). Their results showed a linear increase in the intake when the protein supplementation was increased (*p* < 0.01).

For the production variables (Table 4), the first aspect to highlight is the difference in the average live weights (LW) at the end of the period. Likewise, this difference is reflected in the difference that was presented in the average daily gain, this being two times higher in favor of the MF group. These differences have a strong economic productive implication, with possible negative effects in terms of the weights of the hot carcass [24,58], time, and a delay in the finishing phase of the HF animals. This delay, in turn, has environmental consequences, as the longer the animal has not reached its productive objective (slaughter), the greater its contribution to the emission of GHGs into the environment [59].

The absolute methane emissions (g CH_4_/day) showed no differences when comparing the steers that were consuming the forage diets with contrasting fiber contents (Table 4). In recent years, studies have been carried out comparing the effects of forage diets with different NDF contents on CH_4_ emissions. Primavesi et al. [17] compared the CH_4_ emissions from grazing animals that were supplemented with two sugarcane varieties with contrasting NDF values, reporting that the use of the variety with a lower proportion of NDF reduced the CH_4_ emissions by 30%, relative to the diet with a higher NDF content. Aguerre et al. [60], on the other hand, determined the effect of feeding diets with different forage-to-concentrate ratios (F:C), resulting in a positive correlation between the amount of NDF and the enteric CH_4_ emissions. In these terms, a diet with differences in its NDF contents should show a difference in its enteric methane production; however, this difference was not demonstrated in our study, although there is a positive correlation between the NDF intake and CH_4_ emissions (Figure 2). Nevertheless, there are other studies that did not find differences in the emissions of CH_4_, despite comparing diets of very contrasting qualities. This is the case in the work that was presented by Dini et al. [61], which compared the effect of low- and high-quality pasture under grazing conditions in two seasons (winter–spring); however, no differences in the emissions were observed, despite the difference in qualities; 70% vs. 42% NDF in the winter and 55% vs. 41% NDF in the spring, for low and high quality, respectively.

It appears that the NDF content of a diet alone does not explain its enteric methane (CH_4_) emissions. Different studies have shown that a higher amount of forage intake causes an increase in the fermentation processes in the rumen, and therefore presents a higher CH_4_ emission rate (g/d) [62,63,64,65,66]. Boadi and Wittenberg [21] proposed that consumption predicts 64% of the variations in daily emissions. This is related to the results of our work, as the higher consumption presented by the MF group of steers could influence their emissions, which would show that a diet with a lower percentage of NDF emissions would not be different (Table 4). This is shown in Figure 1 and Figure 2, where a correlation of the CH_4_ emissions is shown with both the DMI and NDF intake, with *r^2^* values of 0.51 and 0.45, respectively. Similar results were presented by Pinares-Patiño et al. [64] and Beauchemin and McGinn [50]. The latter reported a correlation of 0.82 between the DMI and methane emissions, while Pinares-Patiño et al. [64] reported correlation values for the absolute methane emissions, with a DMI of 0.53 and 0.42 with NDFI. Likewise, Alemu et al. [67] performed these regressions on accurate intake and emission measurement techniques, such as GreenFeed (GEM) and respiration chamber (RC). For both techniques, the increment values per kg of the DMI, of 12.3 g (+6%) and 15.2 g (+8.2%) of CH_4_, were presented for GEM and RC, respectively, which was very similar to that which was presented in our study, of 12.8 g CH_4_ (+6.7%). However, Alemu et al. [67] presented correlations of only 0.19 and 0.28 for GEM and RC, respectively. On the other hand, Kurihara et al. [62] reported higher correlation values (0.99) and also reported a positive linear regression, but with an increase of 41 g of CH_4_ per kg of the DM that was ingested (+14%). It is interesting to observe the behavior of these relationships with their respective diets, as they are apparently dependent on the scenario in which they are evaluated, and may be important when developing emission factors and different mitigation strategies [6].

Dall-Orsoletta et al. [68] compared the methane (CH_4_) emissions from dairy cows that were consuming diets with the same NDF content (TMR vs. PMR). A higher intake of NDF (6.6 vs. 6 kg/day, respectively) implied higher emissions (656 vs. 546 g CH_4_/day), even though both diets had a similar digestibility. However, others have pointed out that increasing the digestibility of the diet would reduce the CH_4_ emissions [52,69]. A high rate of fluid passage can reduce archaeal populations, leading to an accumulation of H_2_ and a reduction in CH_4_ emissions, because of the feedback inhibition of the H_2_ production [69,70]. The CH_4_ emissions should be analyzed, while taking into consideration both the intake and the quality of the feed that was supplied [71].

Different studies have indicated that the best way to evaluate methane (CH_4_) emissions is through intensity, especially in relation to intake [72]. In terms of the emission intensity per DMI, the MF group was lower than the HF group (Table 4). With an increasing feeding level, due to a good digestibility and high intake, there would be a higher efficiency of energy consumed, and thus, a lower CH_4_ production relative to the intake [20,61,73,74]. Different results of g CH_4_/kg DMI have been reported. Andrade et al. [75] reported values between 22.9 and 25.4 g, Dini et al. [37] reported values between 21.6 and 22.7 g, and Dini et al. [61] reported values between 17.9 and 20.2 g, which are all intensities that are within the ranges that were found in this study (15.3 to 29.5 g CH_4_/kg DMI). However, Gere et al. [76] reported DMI values between 11.8 and 12.1 g CH_4_/kg, which are well below those that were presented in this work, and is probably due to the higher consumption that was presented in that study. In turn, the emission intensity per kg of the NDF that was ingested did not show significant differences (*p* > 0.05), most likely due to the low difference in the NDF content between the two diets (5%), and therefore, the intake of this component [3]. On the other hand, the emission intensity of the MF animals, which was expressed in relation to the average daily gain (g CH_4_/kg ADG), was almost half that of the HF group (*p* < 0.05). This is much higher than that which was reported by Maciel et al. [5], where the grazing intensity of Nellore and Angus steers was evaluated, which presented with similar ADG values (with a mean ADG intensity value of 129.5 g CH_4_/kg). This can be explained by the worse feed conversion that was presented in our study, which was, on average, 18.1, much higher than the 8.8 average that was presented by Maciel et al. [5], which is possibly due to the supplementation that these animals received.

Methane (CH_4_) production is considered to be an energy loss [16]. The Ym value that was obtained for each of the forage diets that were evaluated (Table 4) confirmed that their quality, which was defined according to the NDF content, can generate major differences in feed use efficiency, and therefore CH_4_ emissions. In this experiment, the MF group presented a Ym value of 6.7%, a figure that was close to the 6.5% +/− 1% that was proposed by the IPCC for animals that are fed forage-based diets. On the other hand, the Ym value that was obtained in the HF diet group (7.5%) is more in line with that which was proposed by the IPCC [40] for animals that are fed low-quality forage diets. Previous studies in Uruguay have determined different Ym values; for grazing dairy cattle of 6.6% [37], while for beef steers in relation to their residual intake (RFI), it was 6.7% for low RFI animals and 9.17% for high RFI animals [77]. Comparing pastures of different quality, Dini et al. [61] reported values of 4.2% to 7.9% for the winter and spring periods, respectively. Studies conducted in Argentina on pasture-fed beef cattle have shown variable values of 4.3% to 8.2% [76].

The analysis of the observed vs. predicted methane (CH_4_) emissions is presented in Figure 3. This analysis allowed for the evaluation of the prediction equations that have been proposed by different works for diets with a high inclusion of forage resources (>60%). The equation that presented the best performance was that of the IPCC 2006 Guidelines, followed by Moraes 2004 (a). Both equations have in common the use of gross energy intake as a component; however, only the second one also included the NDF and BW as components. Escobar-Bahamondes et al. [27] used a high-forage dataset to determine the best fit equations, whereas the International Panel on Climate Change Tier 2 method is presented as the best performing, with an *r^2^* = 0.71 and RMSE of 39.8 g CH_4_/day, which has very similar accuracy but more precision when compared with our study (*r*^2^ = 0.7 and RMSE of 128 g CH_4_/day). The relatively good performance of the IPCC [40] equation for the high forage diets may reflect that the Ym that was used (6.5%) is from a wide range of diets, with a high proportion of forage. However, Ellis et al. [26] demonstrated that the accuracy of the IPCC (2006) Tier 2 methodology is low, and therefore, when used at the farm scale, this approach could lead to an inaccurate estimate of the CH_4_ released. This is probably because the datasets that were used in these papers did not consider the idea of separating by feed type and forage proportion [27]. When used for the cattle that were fed high grain diets, the existing CH_4_ prediction models were generally inaccurate and lacked precision. In the work presented by Benauda et al. [30], the IPCC model performed moderately well, because it did not take into account the differences in dietary lipids, NDF and starch contents, and diet quality effects. In this same work, the author pointed out a particularly important point, which is that the uncertainties and discrepancies that are associated with CH_4_ measurement techniques were expressed as coefficients of variation (CV), which can range from 20 to 27%, depending on which emission variable is considered. In the same vein, Benchaar et al. [63] stated that the accuracy of the models depends on the assumptions of the model, as well as on the accuracy of the input values that are needed to use the model.

Within this analysis, emphasis was placed upon the studies that were mainly conducted in the Americas, principally in South America (58 observations and 18 studies), which showed different results when comparing diets of different qualities and the methane emissions in absolute terms (g CH_4_/day). For example, in Uruguay, Dini et al. [61] and Oscarberro et al. [78] reported grazing average emission values of 109–177 g CH_4_/day and 140–329 g CH_4_/day for heifers, respectively. For grazing lactating cows, Loza et al. [79] reported values between 353 and 374 g CH_4_/day, while in Argentina, for the same animal category and grazing system, Gere et al. [76] reported values from 157 to 203 g CH_4_/day. Studies in Brazil, for example, those by Maciel et al. [5], Oliveira et al. [4], and Hoffman et al. [80], which measured heifers and steers with forage diets, showed results with a wider range of 80–184 g CH_4_/day. All are very similar values to the ranges that were presented in our study (151 and 293 g CH_4_/day), despite being in different categories but using the same forage diets.

It should be noted that many of these studies and measurements were performed on different categories of animals and under different grazing conditions, a factor to consider since our study was performed on housed steers that did not harvest their own feed. Undoubtedly, the measurement of the feed intake of pasture-raised animals is still costly and of a low precision [4,22,23]. This, combined with the errors that are associated with enteric methane production measurement techniques [81,82], can lead us to erroneous conclusions. For this reason, this study, through the confinement of the animals, allowed us to evaluate the effects of a forage diet by controlling the quality and quantity of the diet that was ingested by the animal.

## 5. Conclusions

Supplying a high-quality forage diet with a moderate fiber content not only showed important positive implications for the animal’s performance, but also in the environmental sustainability of the system. This improvement in productivity was also accompanied by a lower intensity of the enteric CH_4_ emissions that were expressed per kg DM intake, per kg of ADG, and per unit of GEI converted into CH_4_ (Ym). When comparing the estimation values of the model equations versus the observed emission values of the diets with a high forage inclusion, despite its relative simplicity, the performance of the IPCC Tier 2 equation is the best, when compared to other, more complex equations that consider diet composition.

## Figures and Tables

**Figure 1 animals-13-01177-f001:**
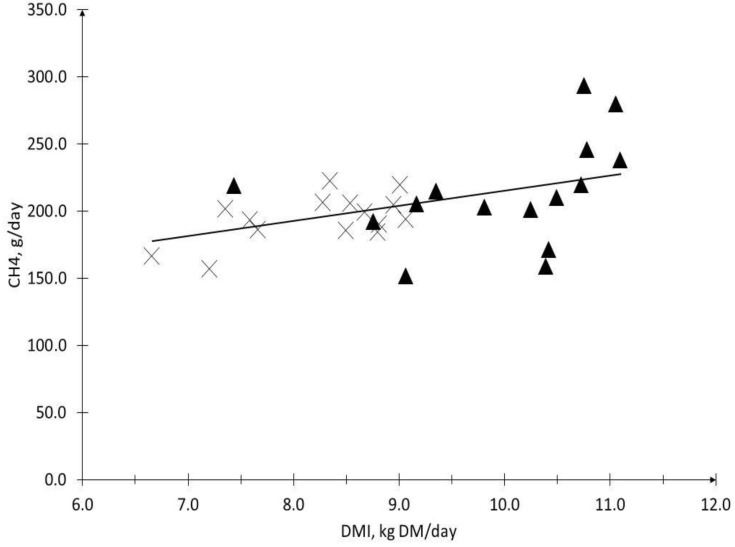
The relationship between dry matter intake (DMI) and CH_4_ production of steers fed moderate-fiber (MF, black triangle) and high-fiber (HF, cross) diets. *r^2^* = 0.51, *p* = 0.0043, and Y = 12.79x + 87.53.

**Figure 2 animals-13-01177-f002:**
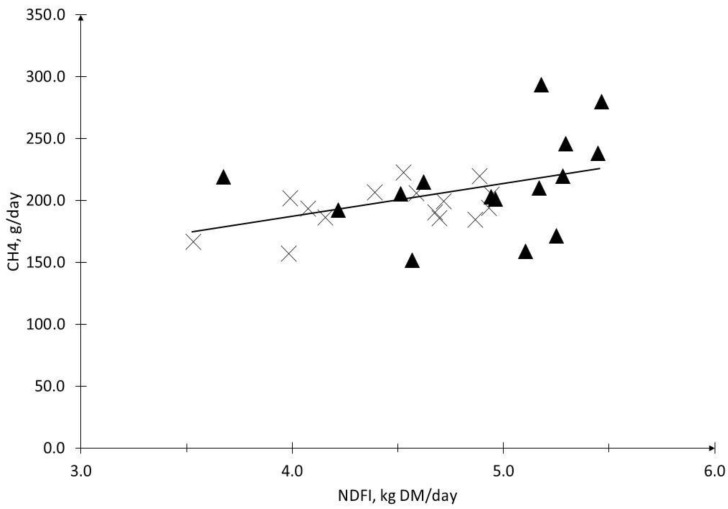
The relationship between neutral detergent fiber intake (NDFI) and CH_4_ production of steers fed moderate-fiber (MF, black triangle) and high-fiber (HF, cross) diets. *r*^2^ = 0.45, *p* = 0.013, and *Y* = 27.52x+ 74.89.

**Figure 3 animals-13-01177-f003:**
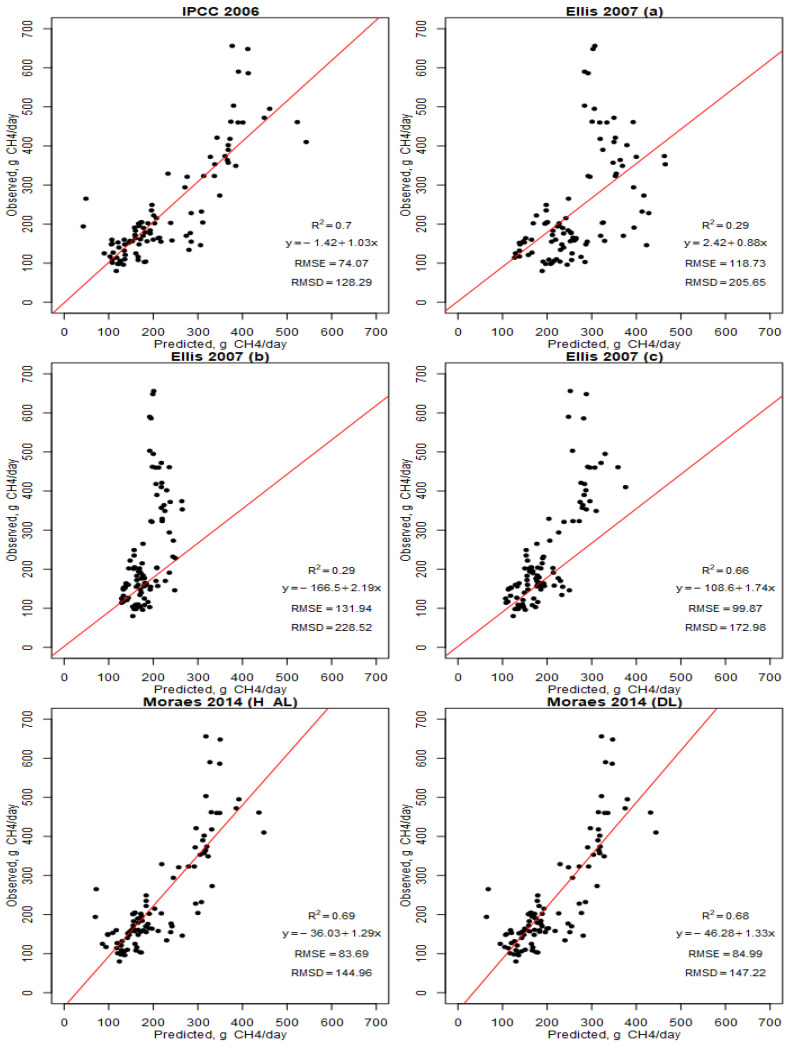
Observed vs. predicted CH_4_ emissions value plots using 97 observations (black points) from 29 different studies in the six models selected. Regression lines are shown in red. RMSE: Root means square error; RMSD: Root means square deviation; and *r*^2^: Pearson’s coefficient.

**Table 1 animals-13-01177-t001:** Chemical composition of forage diets with moderate fiber content (MF) and high fiber content (HF).

	Diet	
Parameter	Moderate Fiber (MF)	High Fiber (HF)
DM, g/kg of fresh matter	579	631
	--------------------g/kg, DM basis—-----------------	
OM	877	880
NDF	491	543
ADF	332	392
Lignin	74	82
CP	148	121
EE	28	24
Ash	123	120
NFC	210	192
NFC:NDF	0.43	0.35

DM: dry matter; OM: organic matter; NDF: neutral detergent fiber; ADF: acid detergent fiber; CP: crude protein; EE: ether extract; and NFC: non-fiber carbohydrates.

**Table 2 animals-13-01177-t002:** List of authors, models, equations and references to estimate CH_4_ emissions evaluated in this study.

Author, Year	Model	Equation	Reference
IPCC, 2006	IPCC 2006	CH_4_ = (0.065 × GEI)/0.05565	[40]
Ellis et al., 2007	Ellis 2007 (a)	CH_4_ = (3.14 + 2.11 × NDFI)/0.05565	[25]
Ellis et al., 2007	Ellis 2007 (b)	CH_4_ = (5.58 + 0.848 × NDFI)/0. 05565	[25]
Ellis et al., 2007	Ellis 2007 (c)	CH_4_ = (−1.02 + 0.681 × DMI + 4.81 × forage)/0.05565	[25]
Moraes et al., 2014	Moraes 2014 (H_AL)	CH_4_ = −1.487 + 0.046 × GEI + 0.032 × (NDF, %) + 0.006 × BW	[29]
Moraes et al., 2014	Moraes 2014(DL)	CH_4_= −0.163 + 0.051 × GEI + 0.038 × (NDF, %)	[29]

CH_4_: enteric methane (g/day); GEI: gross energy intake (MJ/day); NDFI: neutral detergent fiber intake (kg/day); DMI: Dry matter intake (kg/day); forage: forage proportion in the diet (%); NDF: neutral detergent fiber (%); and BW: body weight (kg).

**Table 3 animals-13-01177-t003:** Intake of nutritional components, apparent digestibility, and NDF of steers fed forage diets with moderate fiber content (MF) and high fiber content (HF).

	Diet			
Parameter	Moderate Fiber (MF)	High Fiber (HF)	SEM	*p-*Value
DM, kg/day	9.9	8.2	0.23	<0.001
OM, kg/day	7.5	6.2	0.17	0.001
NDF, kg/day	4.9	4.5	0.12	0.02
ADF, kg/day	3.3	3.2	0.08	0.49
Lignin, kg/day	0.74	0.67	0.02	0.02
CP, kg/day	1.5	1.0	0.03	<0.001
EE, kg/day	0.28	0.20	0.01	<0.001
Ash, kg/day	1.2	1.0	0.03	<0.001
DMD, %	60.7	50.4	1.02	<0.001
NDFD, %	56.6	47.6	1.4	<0.001

DM: dry matter; OM: organic matter; NDF: neutral detergent fiber; ADF: acid detergent fiber; CP: crude protein; EE: ether extract: DMD: dry matter digestibility; and NDFD: neutral detergent fiber digestibility.

**Table 4 animals-13-01177-t004:** Productive and methane emission variables of steers fed forage diets with moderate fiber content (MF) and higher fiber content (HF).

	Diet			
Parameter	Moderate Fiber (MF)	High Fiber (HF)	SEM	*p-*Value
Production				
Initial LW, kg	447	441	9.02	0.62
Final LW, kg	513	472	8.75	0.002
ADG, kg/day	0.65	0.32	0.02	<0.001
Emission				
CH_4_, g/day	214	193	7.56	0.054
CH_4_, g/kg ADG	342	649	48.7	<0.001
CH_4_, g/kg DMI	21.7	23.7	0.64	0.022
CH_4_, g/kg NDF intake	44.09	43.77	1.27	0.97
Ym, %	6.7	7.5	0.2	0.008

LW: live weight; ADG: average daily gain; DMI: dry matter intake; and Ym: methane yield.

## Data Availability

The data presented in this study are available in the article. Further information is available upon request from the corresponding author.

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
