# Peer review of "Beef Steers and Enteric Methane: Reducing Emissions by Managing Forage Diet Fiber Content"

_animals, 2023, doi:10.3390/ani13071177_

Round 1

Reviewer 1 Report

Dear authors,

It was a pleasure to revise your manuscript. This is exceptionally well done. 

Please, correct minor typos regarding CHsubscripts, for example (e.g., line 427). 

Cordially, 

Reviewer 2 Report

This is a study on the effects of dietary fibre content on animal growth and enteric methane emission. Line 55: It is good practice not to start a sentence/paragraph with an abbreviation or chemical symbol. Better just to write methane out in full.

Line 95-96: Do you have a reference for these equations?

Line 166-168: It is not clear how the individual animal was identified. EID ear tag?

Line 169-171: This is not a robust method of calculating ADG. The slope of a  regression line through all recorded weights against time should be used instead, and the related analyses and conclusions revised.

Line 270-271: What were these conclusions and how are they relevant to your analysis?

Line 279: No need to provide p values for insignificant results.

Line 294-297: Your interpretation of p-values is confusing. You say =0.08 is a tendency, but then say that >0.05 is no difference.

General: 

Changing between usage of methane and CH4 (subscript 4's are also different sizes throughout.) please be consistent. Could be more concise in places.

Reviewer 3 Report

The authors conducted a trial on steers to evaluate to methane emission in two different diet groups and to test six models for the estimation of methane emissions.

Main comments on the manuscript:

-Authors call the applied two feeding regimes as high-fiber and low-fiber groups and the NDF content of the diet were 49.1% vs. 54.3%. These are not huge differences in fiber content, that's why I recommend to use "moderate vs. high".

- Authors should take it into consideration in the discussion that although the main focus of the manuscript is about fiber content of the diet, the difference between groups in NDF content is approx. 10% while it is more than 20% for crude protein. 

- The duration of the experiment is revealed only in 2.6 section, I would suggest to write it clearly in the experiment design.

- L37-40: The results confirm that fiber content influences methane emission and followed by the statement about IPCC 2006 model as the most suitable for the estimation of methane emission, but this model is not calculating with the fiber content of the diet. I see that the first case is about is "g methane per kg DMI" and the latter is "g methane per day", but it is a little bit confusing in one sentence. 
